# Assessment of the Influence of Size and Concentration on the Ecotoxicity of Microplastics to Microalgae *Scenedesmus* sp., Bacterium *Pseudomonas putida* and Yeast *Saccharomyces cerevisiae*

**DOI:** 10.3390/polym14061246

**Published:** 2022-03-19

**Authors:** Martina Miloloža, Kristina Bule, Viktorija Prevarić, Matija Cvetnić, Šime Ukić, Tomislav Bolanča, Dajana Kučić Grgić

**Affiliations:** 1Faculty of Chemical Engineering and Technology, University of Zagreb, Marulićev trg 19, 10000 Zagreb, Croatia; miloloza@fkit.hr (M.M.); kbule@fkit.hr (K.B.); vprevaric@fkit.hr (V.P.); mcvetnic@fkit.hr (M.C.); tbolanca@fkit.hr (T.B.); 2Department for Packaging, Recycling and Environmental Protection, University North, 48000 Koprivnica, Croatia

**Keywords:** microplastics, ecotoxicity assessment, size influence, concentration influence

## Abstract

The harmful effects of microplastics are not yet fully revealed. This study tested harmful effects of polyethylene (PE), polypropylene (PP), polystyrene (PS), polyvinyl chloride (PVC), and polyethylene terephthalate (PET) microplastics were tested. Growth inhibition tests were conducted using three microorganisms with different characteristics: *Scenedesmus* sp., *Pseudomonas putida*, and *Saccharomyces cerevisiae*. The growth inhibition test with *Scenedesmus* sp. is relatively widely used, while the tests with *Pseudomonas putida* and *Saccharomyces cerevisiae* were, to our knowledge, applied to microplastics for the first time. The influence of concentration and size of microplastic particles, in the range of 50–1000 mg/L and 200–600 µm, was tested. Determined inhibitions on all three microorganisms confirmed the hazardous potential of the microplastics used. Modeling of the inhibition surface showed the increase in harmfulness with increasing concentration of the microplastics. Particle size showed no effect for *Scenedesmus* with PE, PP and PET, *Pseudomonas putida* with PS, and *Saccharomyces cerevisiae* with PP. In the remaining cases, higher inhibitions followed a decrease in particle size. The exception was *Scenedesmus* sp. with PS, where the lowest inhibitions were obtained at 400 µm. Finally, among the applied tests, the test with *Saccharomyces cerevisiae* proved to be the most sensitive to microplastics.

## 1. Introduction

Plastic pollution has become a serious environmental problem [1,2,3,4]. Nevertheless, the annual world’s production of plastics is continuously increasing. According to the latest report of PlasticsEurope [5], the annual plastic production for the year 2021 was 367 million metric tons, excluding the production of recycled plastics. In addition, the EU Environment Agency presented concerning projections of further production growth to over 25 billion metric tons in 2050 [6]. Looking the type of plastic polymers, polyethylene (PE), polypropylene (PP), polystyrene (PS), polyvinyl chloride (PVC) and polyethylene terephthalate (PET) account for 90% of the world production. Accordingly, these are also the most common types of plastics in the environment [7].

Plastic particles smaller than 5 mm, popularly known as microplastics (MPs), are of particular concern to scientific community [8]. MPs particles have a high hazardous potential. They can affect organisms physically (after ingestion) [9] and chemically (polymer type and chemical composition) [10], and can serve as carriers for various pollutants or as substrates for pathogenic organisms [9]. In addition, MPs particles are chemically very stable and usually non-biodegradable; therefore, they can remain in the environment for hundreds of years. Due to their small size, they are ubiquitous in practically all environments. However, their presence in the aquatic environment is of particular concern. Namely, once these contaminants enter the water, aquatic organisms feed on them, and MPs enter the food chain [11]. In addition, aquatic macrophytes have been reported to retain MPs to a significant extent, increasing thus the contamination in areas covered by aquatic grasses [12,13] and accordingly increasing the exposure of organisms living there.

The harmful nature of MPs particles has not been fully explored, but it is undeniable that there are harmful effects [7,8,14,15]. Therefore, scientists are still intensively conducting toxicity tests on various organisms. Test on the toxicity of MPs often use crustaceans *Daphnia magna* or zebrafish *Danio rerio* as test organisms [16]. Thus, ingested MPs have been reported to accumulate in body of *Daphnia magna* without significant effects on survival and reproduction for particle sizes 63–75 μm [17], while increased mortality and reduced reproductive capacity have been reported for sizes below 5 μm [18,19]. Lei et al. [20] reported that MPs caused intestinal damage in *Danio rerio*. The use of microorganisms in toxicity tests has also become popular: microorganisms are very available, it is easy to cultivate them, and their life-cycle is relatively short which provides fast results. In addition, these tests are simple, inexpensive, and provide accurate results [21]. The most commonly used microorganisms are algae and their growth inhibition is monitored as a toxicity effect [16]. A decrease in chlorophyll content and photosynthetic activity due to lower expression of photosynthetic genes, shading effect, growth inhibition, oxidative stress, physical deterioration of microalgal cells, homoaggregation and heteroaggregation have been identified as the main negative effects for microalgae [22].

Numerous factors can influence the impact of MPs on a selected organism [23]: type, shape, size and concentration of MPs, presence of additives, etc. For example, Tunali et al. [24] studied the impact of different PS concentrations on the growth and chlorophyll content of microalga *Chlorella vulgaris*; the size of PS particles was 0.5 μm. The authors reported maximal harmfulness (28.9% of growth inhibition and 21.3% decrease in chlorophyll content) at the highest concentration tested (1000 mg/L), while no effect was observed at concentrations below 50 mg/L. Zhang et al. [25] investigated the negative effect of PVC MPs on the microalga *Skeletonema costatum* during a 4-day exposure. They tested the effect of concentration at two sizes: 1 µm and 1 mm; the concentration ranges were 0–50 mg/L and 0–2000 mg/L, respectively. Concentrations of 1 µm particles had a significant effect on microalgae: higher concentrations resulted in a stronger negative effect. PVC particles of 1 mm also caused negative effect, but no significant influence of particle concentration was found. Lagarde et al. [26] tested the influence of PP and high-density PE microparticles on the freshwater microalgae *Chlamydomas reinhardtii* and confirmed that the type of MPs played a significant role in the harmfulness of plastic waste. The tested polymers acted similarly during short-term exposure: rapid colonization by *C. reinhardtii* was observed in both cases. However, a difference was reported in case of long-term exposures. In the case of PP, hetero-aggregates appeared after 20 days of contact and their size continued to increase during the experiment. In contrast, no aggregation was observed in case of high-density PE.

The aim of this work was to test the toxicity of 5 of the most-common types of MPs: PE, PP, PS, PVC, and PET. In general, hazardous substances do not have the same effects on all organisms. The effects differ not only in intensity but in mode of action as well. Therefore, in order to obtain more relevant information about the toxicity of the tested MPs, we decided to perform toxicity tests on three different microorganisms: the freshwater microalga *Scenedesmus* sp., the bacterium *Pseudomonas putida*, and the yeast *Saccharomyces cerevisiae*.

## 2. Materials and Methods

### 2.1. Design of the Experiment

In this study, five types of MPs most commonly found in the environment were used: PE, PP, PS, PVC, and PET (Appendix A). Two factors that might influence the harmfulness of MPs were tested: concentration and particle size. Five concentration levels (50, 250, 500, 750, and 1000 mg/L) and three size levels (200, 400, and 600 μm) were combined using a full factorial methodology: i.e., the experiment was designed using all possible combinations of levels of the two factors.

Three different tests for acute toxicity were performed: growth inhibition test with the microalga *Scenedesmus* sp., growth inhibition test with *Pseudomonas putida*, and yeast toxicity test with *Saccharomyces cerevisiae.* The microalga *Scenedesmus* sp. was selected because microalgae are primary producers that play an important role in the food chain [27], the bacterium *Pseudomonas putida* is commonly used as a representative of heterotrophic microorganisms in freshwaters [28], and yeasts, as eukaryotes, are considered good organisms for toxicity evaluation [29]. The selected microorganisms were easy to cultivate, they all had a short life cycle, their sensitivity to various contaminants has already been confirmed, and the cost of their toxicity tests was low.

The growth inhibition test with *Scenedesmus* sp. is relatively common. Bacterial growth inhibition tests are rarely used to determine the toxicity of MPs, and to our knowledge, this is the first study applying *Pseudomonas putida*. Further, we found no report in which *Saccharomyces cerevisiae* was used in determining the toxicity of MPs. Nomura et al. [30] applied such test to determine the toxicity of PS latex nanoparticles, which were far below the size-range tested in our study.

The standard ecotoxicity test with *Scenedesmus* sp. lasts 72 h, whereas the duration of the standard tests with *Pseudomonas putida* and *Saccharomyces cerevisiae* is 16 h. However, the tests with *Pseudomonas putida* and *Saccharomyces cerevisiae* originally refer to solutions and not suspensions, as was the case in our study. Therefore, we performed some preliminary experiments for these two tests to see if the standard methods were applicable. The preliminary experiments showed that CFU did not change within 16 h, which was not the case at 72 h. Therefore, we set an identical contact time of 72 h for all three tests. 

### 2.2. Preparation of Microplastics

Various plastic products were used as a source of MPs: bags for PE, spoons for PP, knives for PS, packaging boxes for PVC, and bottles for PET. These products were purchased from the store. The products were first cut into smaller pieces with scissors and then ground in a cryo-mill (Retsch, Haan, Germany) with liquid nitrogen to improve the grinding. The ground particles were air-dried at room temperature for 48 h; afterwards they were sieved on stainless steel screens to obtain size classes: 100–300, 300–500, and 500–700 µm. The sieved MPs particles were stored in glass bottles. The Attenuated Total Reflectance Fourier Transform Infrared (ATR-FTIR) spectroscopic analysis (Spectrum One, Perkin Elmer, Waltham, MA, USA) was performed to verify the type of the plastics. The characteristic ATR-FTIR spectra are shown in Appendix A.

An appropriate amount of MPs particles (considering the final MPs concentrations of 50, 250, 500, 750, and 1000 mg/L) was placed in a glass flask before conducting the toxicity experiments. The flasks were filled with 70% ethanol and shaken on a rotary shaker (Unimax 1010, Heidolph, Schwabach, Germany) at 160 rpm for 10 min to sterilize the MPs particles. The sterilized particles were separated from the ethanol suspension by vacuum membrane filtration using cellulose nitrate 0.45 µm sterile filters (ReliaDisc^TM^, Ahlstrom-Munksjö, Helsinki, Finland) and washed with sterile deionized water. Finally, the particles were quantitatively transferred into sterile flasks for the microalgae and bacteria or sterile bottles for the yeast. The transfer was performed using sterile technique.

### 2.3. Preparation of Test Microorganisms

Microalgae *Scenedesmus* sp. (Appendix A) were obtained from the Ruđer Bošković Institute (Zagreb, Croatia). *Scenedesmus* sp. was activated in sterilized liquid basal medium at 25 ± 2 °C for 12/12 h light/dark cycle. Sedimentation of microalgae was prevented by aeration through a 0.45 µm sterile filter (ReliaDisc^TM^, Ahlstrom-Munksjö, Helsinki, Finland). The number of live algal cells (shown as *Colony Forming Unit*, CFU) was determined using an optical microscope (Olympus BX50, Olympus Optical, Tokyo, Japan) with Thoma counting chamber. If necessary, the suspension was diluted to the initial number of 10^5^ cells/mL.

Bacterium *Pseudomonas putida* (Appendix A) was cultivated according to the guidelines [31]. Prior to the experiment, the culture was additionally cultivated in mineral media during 5 ± 0.5 h [32] on a rotary shaker at 160 rpm and room temperature. The optical density of the bacterial suspension was measured at 436 nm using a spectrophotometer DR/2400 (Hach, Loveland, CA, USA) and the initial value was set to 0.2 by diluting the suspension.

Yeast *Saccharomyces cerevisiae* (Appendix A) was cultivated on yeast medium agar (3 g/L of yeast extract, 3 g/L of malt extract, 5 g/L of peptone, 10 g/L of glucose, and 15 g/L of agar; pH value was 7.0 ± 0.2) for 10–12 h at 30 ± 0.1 °C. The yeast suspension was prepared in sterile deionized water, and the initial optical density was adjusted to an absorbance value of 3.0, which was measured at 550 nm using a spectrophotometer DR/2400 (Hach, Loveland, CA, USA).

### 2.4. Toxicity Tests

The algal growth inhibition test with the freshwater microalga *Scenedesmus* sp. was performed according to OECD guidelines [33,34]. The test was performed in 250 mL sterile Erlenmeyer flasks on a rotary shaker at 160 rpm and 23 ± 2 °C; the working volume was 100 mL. The working flasks contained suspension of algae, basal medium, and MPs particles. The test required a control flask, which was used for comparison. The control flask did not contain MPs particles, only the algal suspension and basal medium only. The initial concentration of dissolved oxygen was 8.65 ± 0.24 mg/L, pH was 8.08 ± 0.17, and the initial CFU value was 4.2 × 10^5^ cells/mL.

ISO guideline 10712 [31] was applied to perform the bacterial growth inhibition test with *Pseudomonas putida.* The experiments were performed in 100 mL sterile Erlenmeyer flasks on a rotary shaker at 160 rpm and 23 ± 2 °C. The working volume was 25 mL. The working flasks contained suspension of bacterium, a mineral medium (composition according to the guideline), and MPs particles. Control flasks were prepared analogously to the previously described algal test. Initial experimental conditions included dissolved oxygen at a concentration of 8.08 ± 0.22 mg/L, pH of 7.04 ± 0.90 and 5.4·× 10^6^ cells/mL. The CFU values were determined each day during the experiments, for both the algal and bacterial tests. The value of the third-day was used to calculate the growth inhibition (INH) according to Equation (1).
(1)$$\mathrm{INH}=\frac{\log{\mathrm{CFU}}_{\begin{array}{l} \mathrm{CONTROL}\\ \mathrm{FLASK} \end{array}}-\log{\mathrm{CFU}}_{\begin{array}{l} \mathrm{WORKING}\\ \mathrm{FLASK} \end{array}}}{\log{\mathrm{CFU}}_{\begin{array}{l} \mathrm{CONTROL}\\ \mathrm{FLASK} \end{array}}}\cdot 100\%$$

Yeast toxicity test was based on the inhibition of saccharose fermentation by *Saccharomyces cerevisiae* [35]. The experiments were performed in hermetically sealed sterile glass bottles (working volume of 30 mL) at 28.0 ± 0.1 °C. The working bottles contained 0.6 mL of yeast suspension, 5 mL of liquid medium (composition according to Hrenovic et al. [36]), and MPs particles. Control bottles were used as well. CO_2_ gas is formed during the saccharose fermentation which increases the pressure in the bottle. Therefore, a syringe was inserted through the bottle cap to collect the fluid (liquid and/or produced CO_2_ gas) pressed from the bottle. The pressed volume was equal to the volume of CO_2_ produced. The volume was collected daily and the three-day cumulative value was used to calculate the inhibition (INH) according to Equation (2).
(2)$$\mathrm{INH}=\frac{V_{\mathrm{CONTROL}}-V_{\mathrm{SAMPLE}}}{V_{\mathrm{CONTROL}}}\cdot 100\%$$

### 2.5. Response Surface Modeling

Modeling of the response (i.e., inhibition) surface was applied to define the influence of concentration (*γ*) and size (*x*) of MPs particles on the growth inhibition (INH) of each test organism. For this purpose, the size intervals of MPs were replaced by corresponding average values: 200, 400, and 600 µm. Three regression models of different complexities were applied to describe the response surface. The models were presented by Equations (3)–(5).
(3)$$\mathrm{INH}=a_{0}+a_{1}\gamma +a_{2}x$$
(4)$$\mathrm{INH}=a_{0}+a_{1}\gamma +a_{2}x+a_{3}\gamma^{2}+a_{4}x^{2}$$
(5)$$\mathrm{INH}=a_{0}+a_{1}\gamma +a_{2}x+a_{3}\gamma^{2}+a_{4}x^{2}+a_{5}\gamma x$$

The letter *a* used in these models stands for models coefficients. MODEL I (Equation (3)) contains linear contributions of the concentration and the particle size, while MODEL II (Equation (4)) is actually MODEL I extended by two quadratic terms. It is expected that first- or second-order polynomials should be adequate for description of dependent variable in most cases involving two independent variables [37]. In order to enclose eventual joint activity of concentration and particle size, MODEL III (Equation (5)) with the interaction term (*γ*∙*x*) was also used.

Calculations and analyses were performed using MATLAB R2010b software (MathWorks^®^, Natick, MA, USA).

## 3. Results and Discussion

Table 1 shows the experimentally determined inhibition values for predefined MPs sizes and concentrations. To statistically determine whether the concentration or the size had a significant influence on MPs harmfulness within the experimental range, the inhibition data were fitted by one linear (Equation (3)) and two polynomial regression models (Equations (4) and (5)). Statistical analysis was performed for each model applied and the results were presented in Table 1, Table 2 and Table 3. Calculations were done with 95% confidence, i.e., the significance level was 0.05. The applied regression models were compared based on related *R*^2^_adj_ values, and the best model for each combination of applied test organism and MPs type was selected. *R*^2^_adj_ was given primacy in front of *R*^2^ because it is considered superior in cases where models with different number of terms need to be compared [38]. The best models were projected at Figure 1, Figure 2 and Figure 3. Based on the best models, conclusions were drawn about the influence of MPs size and concentration.

### 3.1. Inhibition of Scenedesmus sp.

Experimentally determined values of growth inhibition of *Scenedesmus* sp. showed a general trend for all five plastics studied: higher concentrations resulted in more intensive inhibitions (Figure 1). No trend was evident for size variations, except perhaps for PS (Figure 1C). However, the highest inhibition values for all five plastics (Figure 1 and Table 1) were found at the lowest particle size (200 μm) and the highest concentration (1000 mg/L), suggesting that the size has an impact on the algal growth, at least at higher MPs concentrations. To check these observations, we fitted the inhibition data with three regression models (Equations (3)–(5)) and performed statistical analysis. The results are presented in Table 2. All applied models proved to be significant in describing the of variability of the dependent variable (i.e., inhibition of algal growth). Namely, high *F*-values and *p*-values below the predefined significance of 0.05 were obtained for all applied models (Table 2).

The linear model (MODEL I), applied for PE experiments, successfully described 95.35% of the variance of the inhibition (*R*^2^ = 0.9535), indicating the existence of a high influence of at least one of the examined factors on the algal growth (Table 2). And indeed, the confidence intervals of the estimated model-coefficients for both coefficients related to the independent variables, did not contain the value 0 and the corresponding *p*-values were below the predefined significance level. This implied that both factors studied: the concentration and the particle size have a significant influence on algal growth. The introduction of quadratic terms in MODEL II resulted in a better fit (higher *R*^2^_adj_ values) and demonstrated the superiority of the model over MODEL I. The analysis of variance performed for MODEL II refuted the conclusion derived from MODEL I and revealed PE concentration as the only influential factor. The coefficient of the size-related quadratic-term had a *p*-value at the boundary of predefined significance. However, the associated confidence interval had a value 0 included, which gave additional conformation that size was not an influential factor. MODEL III, which had the best fit (*R*^2^_adj_ = 0.9502), confirmed this statement. In addition, the inhibition surface described by MODEL III (Figure 1A) clearly showed that higher PE concentrations were more harmfull for the algae. For the lowest concentration tested (50 mg/L), inhibitions of 3.61% or less were obtained (Figure 1A). Accordingly, it can be assumed that the concentration of PE has no significant effect at lower concentration levels. This is in agreement with report of Garrido et al. [39] who performed a similar experiment, but with the microalga *Isochrysis galbana*. Garrido et al. conducted the experiment using smaller PE particles (up to 22 μm) and lower concentrations (up to 25 mg/L) and found no harmful effects on the algae. The exposure time was identical to that in our study.

Somewhat lower values of the determination coefficients (*R*^2^, *R*^2^_adj_) were observed while modeling the inhibition surface in the PP experiment (Table 2), although all three applied models were still significant. Though the best model was MODEL III, all three models gave an identical conclusion that, within the experimental range, the concentration of PP microparticles had a significant influence on the algal growth while changes in the particle size had no statistically significant effect. The influence of PP concentration is more expressed at higher concentrations (Figure 1B).

Application of MODEL I in the case of the PS experiment resulted in generally lowest percentage of the explained inhibition variance (*R*^2^ = 0.8570; Table 2). However, the introduction of quadratic terms (MODEL II) greatly improved fitting of the inhibition data: the increase in *R*^2^_adj_ was 0.1003. Both these models indicated concentration and size of PS microparticles as influential factors in *Scenedesmus* sp. inhibition. Inclusion of the interaction term in the regression model (MODEL III) did not bring additional benefit to the fitting, indicating lack of joint concentration-size activity of PS microparticles. The inhibition surface described by MODEL II is presented in Figure 1C. The surface clearly shows that higher PS concentrations are associated with higher inhibition values. For the size of PS, the surface shows the lowest inhibition for 400 µm particles. The reason why larger and smaller particles had a higher influence on the inhibition could be due to the different inhibition mechanisms. Namely, Bhattacharya et al. [40] reported that one of the reasons for inhibition of the algal growth was adsorption of the charged PS beads on the cell, which obstructed algal photosynthesis. There are two very likely types of the obstruction. The first one is a physical blockage of gas-transfer through the cell membrane (reduced uptake of CO_2_) which should be more expressed for smaller particles due to more complete coverage of the cell surface. The second is a shading effect. It seems important to point out that, of the five tested MPs polymers, four were transparent while only PS had white nontransparent particles. Adsorption of such particles on the cell surface reduces the amount of light which is essential for the normal functioning of photosynthetic organisms. The shading effect should be more evident when larger particles are adsorbed. Furthermore, Besseling et al. [19] performed an experiment very similar to that described in this chapter, but for PE nanoparticles with an exact size of 0.070 μm. The test organism was the microalga *Scenedesmus obliquus*, the exposure time was the same as in our study, and the concentration interval (44–1100 mg/L) was almost identical to ours. Besseling et al. reported that increasing PS concentration resulted in increased inhibition of algal growth. Despite the fact that, based on MODEL III shape (Figure 1C), we expected higher values of inhibition in nano-size area, Besseling et al. reported much lower inhibition values. For example, at a concentration of 1000 mg/L, they achieved an inhibition of about 2.5%, whereas in our study 11.34% was obtained experimentally at the same concentration (for 200 μm-sized particles). During the experiment, Besseling et al. monitored not only the inhibition of algal growth but the level of Chlorophyll A in the algal cells as well. From the observed changes in the level of Chlorophyll A, they concluded that in the case of nano-size PS particles additional mechanism must be included in obstruction of *Scenedesmus* sp. photosynthesis, beside the previously mentioned blockage of CO_2_ transfer through the cell membrane.

Interesting results were obtained in the regression analysis in the case of the PVC experiment (Table 2). Namely, MODEL I and III pointed to concentration and size of PVC microparticles as statistically influential factors, while MODEL II recognized only the concentration. Comparison of *R*^2^_adj_ values showed the reason for this inconsistency. *R*^2^_adj_ value of MODEL II was lower than that of MODEL I. This proved that the introduction of quadratic terms was insignificant for describing the inhibition data and made MODEL II inferior to MODEL I. In contrast, introduction of the interaction term into the regression model (MODEL III) was very beneficial for data fitting and made MODEL III the best option for the regression. According to MODEL III (Figure 1D), at the lowest concentration (50 mg/L), there is a very slight variation in inhibition values within the size-range of 200-600 μm. However, as the PVC concentration increases, the effect of the particle size variation becomes more pronounced, showing increasing harmfulness of the smaller PVC microparticles. This behavior corresponds with some similar studies [41,42] which reported higher toxicity of smaller PVC particles on algae.

MODEL I indicated a significant influence of the concentration and particle size in the case of the PET experiment. However, analysis of the other two models, which were found to be superior to MODEL I, showed that this was not the case. These two models only recognized significant influence of PET concentration. The best fitting model to describe the experimental data was MODEL III, as it gave the in best data-fit (*R*^2^_adj_ = 0.8965). Comparison of the entire set of data presented for *Scenedesmus* sp. (Figure 1) showed that, within the experimental range, *Scenedesmus* sp. was least sensitive to PET as the exposure resulted in the lowest inhibition values. Also, the increase in inhibition with the increase in MPs concentration was least expressed in the case of PET. Recent researches [43,44,45] have confirmed that some microalgae can produce PET hydrolyzing enzymes called PETases and use PET as substrate. Although it is difficult to claim that this was the case in our experiment without performing a detailed analysis, especially when dealing with a 3-day exposure period, this assumption cannot be dismissed. For example, Moog et al. [43] exposed the photosynthetic microalga *Phaeodactylum tricornutum* to PET and observed a progressive increase in the concentrations of mono(2-hydroxyethyl) terephthalic acid and terephthalic acid, which are the main products of the PET hydrolysis, after only 3 days of exposure.

### 3.2. Inhibition of Pseudomonas putida

Exposure of the bacterium *Pseudomonas putida* to MPs particles in predefined ranges of the concentrations and particle sizes resulted in inhibition values (Figure 2) similar to those obtained for *Scenedesmus* sp. Again, higher values of maximal inhibition were determined in the PE, PS, and PVC experiments (12.35, 14.74, and 11.89%, respectively; Table 1) than in the PP and PET cases (8.48 and 7.42%, respectively). Furthermore, these maximal inhibitions were all determined at maximal MPs concentration (1000 mg/L) and the minimal size (200 μm), suggesting the potential influence of the concentration and the size of MPs particles on *Pseudomonas putida* growth. The influence of MPs concentration became even more evident after analyzing all experimental data, as an apparent trend of the inhibition increase with the increase in the concentration existed for all five applied MPs (Table 1). The influence of the particle size was not so obvious (similar to the case of *Scenedesmus* sp.), except perhaps in the cases of PVC and PET, where it appeared that smaller MPs sizes resulted in higher inhibitions. To confirm or refute all these observations, we modeled the inhibition surface (Equations (3)–(5)) and performed a statistical analysis. The results of the analysis are presented in Table 3.

Exposure of *Pseudomonas putida* to PE resulted in relatively high inhibitions (only PP resulted in higher inhibitions; Figure 2). All three regression models applied to describe the inhibition surface were statistically significant (*p*-values of the models were less than 0.05; Table 3), although some of them presented rather poor correlation (MODEL I). MODEL I and MODEL III recognized both tested factors, PE concentration and particle size, as statistically influential, while MODEL II recognized the concentration only. However, MODEL II had the lowest *R*^2^_adj_ values, indicating its inferiority to MODEL I and especially MODEL III. Obviously, the inclusion of the quadratic terms in the case of MODEL II was not beneficial for describing the response surface. In contrast the inclusion of the interaction term (MODEL III) overcame issues of MODEL II and resulted in a huge improvement in *R*^2^_adj_ value; consequently, MODEL III provided the best fit of the experimental data. The significance of the interaction term, i.e., of the joint concentration-size activity, was confirmed by the calculated *p*-value for the coefficient *a*_5_ (*p* = 0.01) which was far below the predefined significance level. The inhibition surface plotted by MODEL III (Figure 2A) showed an interesting behavior of the inhibition. Namely, at the lowest concentration (50 mg/L), inhibition values decreased with reduction in the particle size up to 400 μm and afterwards it stagnated. However, at higher concentrations, the inhibition increased with the reduction of the particle size. Increasing of the concentration resulted in increasing of the inhibition, as observed previously.

MODEL I provided the best fit of the experimental data in the case of *Pseudomonas putida* exposure to PP microparticles (Table 3). Analysis of the model showed PP concentration and size as significant factors that affected bacterial growth. The introduction of quadratic terms (MODEL II) and the interaction term (MODEL III) reduced the suitability of the regression models to describe the inhibition surface. Graphical presentation of the best model (Figure 2B) showed that negative influence of PP microparticles on *Pseudomonas putida* increased with higher concentrations and smaller particles sizes.

Statistically, the concentration of microparticles was the only factor that had an influence of *Pseudomonas putida* growth in the PS experiment (Table 3). MODEL I also indicated the influence of the size, but this model was inferior to MODEL II and MODEL III. MODEL III proved to be the best choice for description of the inhibition surface. Inclusion of the interaction factor in the case of MODEL III improved *R*^2^_adj_ value for 0.0051; however, no significance of the interaction term was found (*p* = 0.17). The model confirmed that an increase in inhibition followed an increase in the concentration (Figure 2C). Exposure of *Pseudomonas putida* to PS microparticles resulted in the highest inhibitions (Figure 2) among the five MPs used. This is most likely related to the styrene contained in PS structure (Appendix A). Namely, styrene is an aromatic hydrophobic compound, known to be a toxic pollutant that can cause negative effects on bacterial cells [46].

All three models applied in the case of PVC pointed to concentration and size as the influential factors. According to MODEL II, the introduction of quadratic term associated with particle size was particularly beneficial for fitting the inhibition values. This is consistent with the parabolic nature of the dependence between inhibition and the particle size seen in Figure 2D. The quadratic term associated with concentration was found to be insignificant. The best model to describe the inhibition surface was MODEL III, whose analysis showed a significance influence of the interaction term (*p*-value for *a*_5_ was less than 0.05) and, accordingly, of the joint concentration-size harmful activity. Giacomucci et al. [47] reported that the bacteria *Pseudomonas citronellolis* are able to degrade PVC, and based on the reported CFU values, it is clear that biodegradation started very early, practically at the 3^rd^ day of exposure. Therefore, it cannot be excluded that the slightly lower inhibition levels of *Pseudomonas putida* that we observed during PVC exposure were due to the onset of PVC biodegradation. However, this remains to be confirmed by future studies.

A very similar conclusion can be given for the PET experiment, as all three models pointed concentration and particle size as influential factors. Again, the inclusion of the quadratic term was significant for the size and not for the concentration. Finally, MODEL III was the best choice for describing the inhibition surface although no significant joint concentration-size activity was confirmed in this case. Based on the structure of the PET polymer, which contains a benzoic ring (Appendix A), we expected slightly higher inhibition values than those obtained. It was known that *Pseudomonas putida* is able to metabolize ethylene glycol [48], which is one of the products of PET depolymerization [49]. Jet, although several other bacteria of the genus Pseudomonas are capable of producing PETase [50,51,52], to our knowledge there are no reports confirming this ability for *Pseudomonas putida*.

### 3.3. Inhibition of Saccharomyces cerevisiae

Toxicity experiments performed with the yeast *Saccharomyces cerevisiae* resulted in much higher inhibition values compared to experiments with *Scenedesmus* sp. and *Pseudomonas putida* (Table 1). Complete or almost complete inhibitions were obtained for four of the five MPs tested: PE, PS, PVC, and PET, indicating a high sensitivity of *Saccharomyces cerevisiae* to the presence of the selected plastic materials. These maximal values were all obtained at maximal MPs concentration (1000 mg/L) and minimal MPs size (200 μm). Therefore, it was not surprising that inhibition of *Saccharomyces cerevisiae*, considering the changes of the concentration, followed the same trend as the inhibitions of *Scenedesmus* sp. and *Pseudomonas putida*: higher MPs concentrations resulted in higher inhibitions. However, unlike the experiments performed with *Scenedesmus* sp. and *Pseudomonas putida*, there also appears to be a clear trend with respect to MPs size, with higher inhibitions obtained for smaller MPs particles. We determined whether this was true by analyzing the applied regression models (Equations (3)–(5)).

Concerning the PE experiment, all three applied models resulted in high *R*^2^_adj_ and *F*-values (Table 4), confirming a good selection of regression models. For example, MODEL I, the simplest of the models used, described 94.49% of the inhibition variance; this means that only 5.51% of the variance remained unexplained by the model. MODEL III, which gave the highest *R*^2^_adj_ value and, accordingly, best described the inhibition of the yeast growth, implied that PE concentration and size were both the influential factors. In addition, the other two models implied the same conclusion. Finally, no joint concentration-size activity was found to be significant because the *p*-value for the interaction term (MODEL III, coefficient *a*_5_) was 0.09. The best model was plotted at Figure 3A. The model confirmed that concentration and size followed the trends assumed by comparing the inhibition data: higher concentrations and smaller sizes of PS microparticles favored the inhibition.

Within the experimental range, exposure to PP microparticles did not result in complete inhibition: maximal value of 80.95% was reached for 200 μm particles at a concentration of 1000 mg/L. Application of the regression models in PP experiments (Table 4) resulted in the lowest values of the coefficients of determination (*R*^2^, *R*^2^_adj_), implying a slightly worse fit of the inhibition data. Nevertheless, all three models proved their significance. MODEL I was quite inferior to MODEL II and especially MODEL III. Therefore, we ignored the idea coming from MODEL I that both factors studied had a significant influence and concluded that only the concentration is influential within the selected experimental range (MODEL II and MODEL III). Although MODEL III provided the best description of the inhibition data, no significant influence of the joint concentration-size activity was statistically confirmed (*p*-value for *a*_5_ was 0.07). The inhibition surface, obtained by the best model (Figure 3B), shows that above 500 mg/L the surface loses its slope and is almost parallel to the xy-plane. Therefore, we assume that PP concentrations higher than 1000 mg/L do not additionally affect the inhibition of the yeast growth.

The quality of the fitting of the inhibition data in the PS experiment increased with the complexity of the model applied (Table 4). However, the improvement in the case of MODEL II compared with MODEL I was rather small (*R*^2^_adj_ increase was 0.0032 only) and none of the quadratic terms included in the model proved to be statistically significant, calling into question the superiority of MODEL II. MODEL III, which had the best *R*^2^_adj_ value, also found quadratic terms to be insignificant. Based on the analysis of MODEL III, the concentration and size of PS microparticles were found to be influential in the experimental range. Influence of the particle size was manifested through the interaction term, i.e., through the joint concentration-size activity. The plotted inhibition surface (Figure 3C) confirmed the previously mentioned observation that inhibition increases with higher PS concentrations.

Analysis of the regression models used in the PVC experiment led to the consistent conclusion that both tested factors had a significant influence on the yeast growth. The best model was MODEL III; the model indicated no significant joint concentration-size activity of PVC particles. Analysis of the inhibition surface (Figure 3D), plotted using the best regression model, showed a reduction (at high concentrations) or stagnation (at low concentrations) in yeast inhibition as particle size was decreased from 600 to 400 μm. For particle sizes below 400 μm, there was a rapid increase in the inhibition.

In the case of PET, the inclusion of quadratic terms (MODEL II) improved the fit of the inhibition data. This refers to the square of the concentration, since the analysis of the model did not confirm the significance of the square of the particle size. However, MODEL III as the best model in the PET experiment showed that both quadratic terms were significant. In addition, the analysis of the model implied significance of the interaction term. The inhibition surface described by MODEL III is shown in Figure 3E.

## 4. Conclusions

The acute toxicity of five MPs: PE, PP, PS, PVC and PET, was determined for the microalga *Scenedesmus* sp., the bacterium *Pseudomonas putida* and the yeast *Saccharomyces cerevisiae*. The monitored toxicity effect was the inhibition of the growth of microorganism. The influence of size and concentration of MPs on inhibition was tested.

Within the experimental range (200–600 µm and 50–1000 mg/L), the maximum value of experimentally determined inhibition for each microorganism/MPs combination was obtained at the lowest particle size and maximum concentration. These values follow further sequences: PE > PVC > PS > PP > PET for *Scenedesmus* sp.; PS > PE > PVC > PP > PET *Pseudomonas putida*; and PS = PVC > PE > PET > PP for *Saccharomyces cerevisiae*. Among the three toxicity tests used, the *Saccharomyces cerevisiae* test proved to be the most sensitive to MPs.

The concentration of MPs proved to have a significant influence on inhibition of all three organisms: higher concentrations resulted in higher inhibitions for all five MPs used. The shape of the inhibition surface for *Saccharomyces cerevisiae* exposure to PP suggests that PP concentrations above 1000 mg/L do not contribute to a further increase in inhibition.

The influence of MPs size was not statistically confirmed in all cases. These exceptions were: exposure of *Scenedesmus* sp. to PE, PP, and PET, exposure of *Pseudomonas putida* to PS, and exposure of *Saccharomyces cerevisiae* to PP. In most cases where a size effect was demonstrated, the inhibition increased as the particles became smaller. A parabolic inhibition-size dependence observed for *Scenedesmus* sp. exposed to PS implies that different inhibition mechanisms prevail at sizes above and below 400 µm.

MODEL I (the model with linear-contributions of concentration and size only) was best for describing inhibition of *Pseudomonas putida* when exposed to PP. In all other cases, MODEL II (model with quadratic terms), and especially MODEL III, were superior to MODEL I. Despite the fact that MODEL III contained the interaction term, the joint concentration-size influence on the inhibition was statistically confirmed in only 6 cases: *Scenedesmus* sp. with PVC, *Pseudomonas putida* with PE and PVC, and *Saccharomyces cerevisiae* with PE, PS and PET.

The observed inhibitions of the microorganisms confirmed the high hazardous potential of 200–600 μm MPs particles when present at concentrations of 50–1000 mg/L. The information on inhibition trends may be an indicator of possible mechanisms of harmful activity. However, additional experiments are required to reveal the true nature of the harmful effects of MPs.

## Figures and Tables

**Figure 1 polymers-14-01246-f001:**
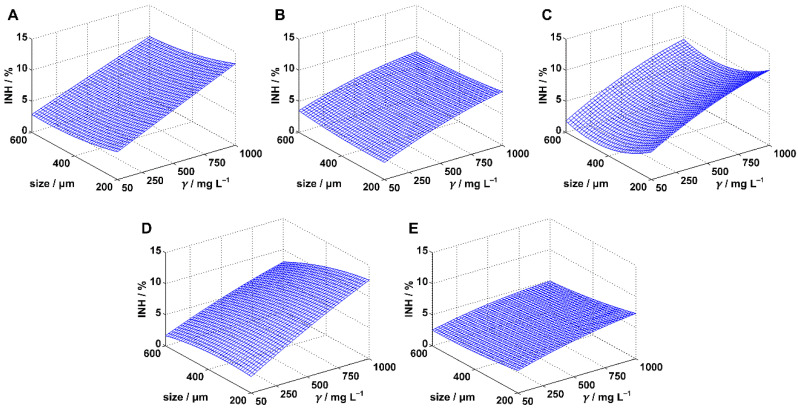
Inhibition surfaces estimated for *Scenedesmus* sp. by the best regression models. The cases are: (**A**) polyethylene, (**B**) polypropylene, (**C**) polystyrene, (**D**) polyvinyl chloride, and (**E**) polyethylene terephthalate.

**Figure 2 polymers-14-01246-f002:**
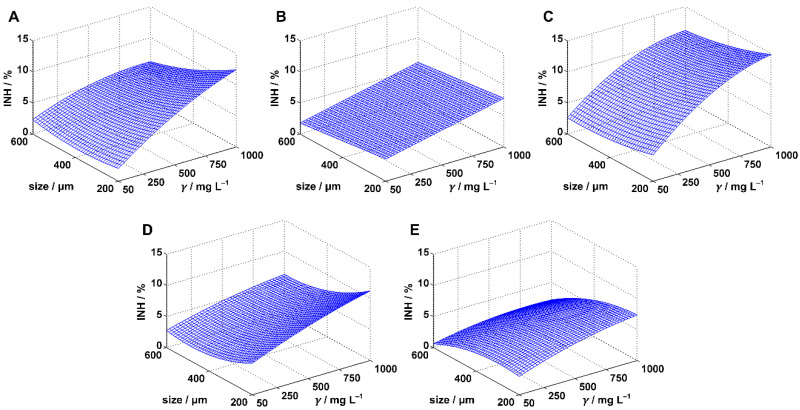
Inhibition surfaces estimated for *Pseudomonas putida* by the best regression models. The cases are: (**A**) polyethylene, (**B**) polypropylene, (**C**) polystyrene, (**D**) polyvinyl chloride, and (**E**) polyethylene terephthalate.

**Figure 3 polymers-14-01246-f003:**
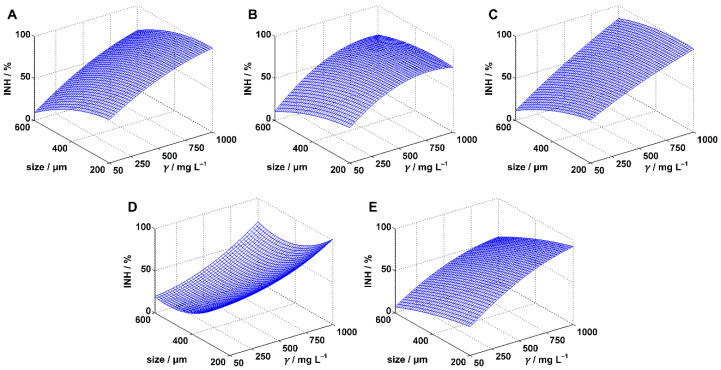
Inhibition surfaces estimated for *Saccharomyces cerevisiae* by the best regression models. The cases are: (**A**) polyethylene, (**B**) polypropylene, (**C**) polystyrene, (**D**) polyvinyl chloride, and (**E**) polyethylene terephthalate.

**Table 1 polymers-14-01246-t001:** Experimentally determined inhibition values (INH) for polyethylene (PE), polypropylene (PP), polystyrene (PS), polyvinyl chloride (PVC), and polyethylene terephthalate (PET) microplastics.

Size/μm	Conc./mg/L	*Scenedesmus* sp.					*Pseudomonas putida*					*Saccharomyces cerevisiae*				
		PE	PP	PS	PVC	PET	PE	PP	PS	PVC	PET	PE	PP	PS	PVC	PET
		INH/%					INH/%					INH/%				
200	50	3.61	3.03	4.56	2.13	4.11	1.87	2.57	3.62	4.49	2.75	49.63	42.86	50.00	63.24	35.24
	250	7.24	4.03	6.27	5.16	4.51	5.04	4.97	8.70	7.35	4.88	65.63	61.86	64.29	66.67	52.13
	500	8.08	6.55	9.26	8.43	5.89	6.53	5.65	11.60	8.56	5.31	80.31	76.29	69.05	73.33	68.09
	750	11.41	6.74	11.34	10.50	5.89	12.19	6.31	12.90	9.00	6.48	98.44	73.20	95.24	83.09	88.10
	1000	12.99	9.13	11.34	12.10	7.63	12.35	8.48	14.74	11.89	7.42	98.76	80.95	100.00	100.00	91.30
400	50	3.61	2.98	2.49	3.10	2.43	1.35	2.33	2.90	2.83	1.73	39.06	25.77	41.62	20.59	20.83
	250	4.93	4.62	3.27	4.13	3.47	5.37	4.57	4.87	4.35	4.99	58.44	54.76	52.38	32.35	43.48
	500	6.97	5.18	4.56	6.54	4.51	5.85	4.66	10.39	5.37	5.52	73.44	74.23	66.67	35.90	58.33
	750	7.81	6.55	6.94	10.23	5.89	6.67	5.23	11.60	5.83	5.30	79.06	75.26	88.10	42.86	68.64
	1000	11.15	8.22	9.26	10.5	5.89	8.00	7.17	12.49	7.37	5.72	90.63	76.19	88.59	76.92	76.19
600	50	2.73	2.70	0.72	2.24	2.43	1.54	1.93	3.10	2.33	1.02	7.810	15.46	9.520	16.67	7.450
	250	4.36	5.18	4.14	3.10	3.47	4.52	3.10	5.85	3.77	1.45	29.06	24.74	26.19	30.95	23.85
	500	5.75	6.74	6.27	4.13	4.51	5.03	3.65	8.10	5.52	1.52	40.48	57.73	54.76	33.33	37.23
	750	8.76	6.95	9.26	6.54	4.51	5.26	4.76	9.75	5.33	1.52	61.54	62.89	64.29	46.67	47.14
	1000	9.86	6.95	9.26	8.43	4.51	7.03	5.56	11.60	6.44	2.84	73.44	64.29	88.10	72.06	54.76

**Table 2 polymers-14-01246-t002:** Statistical analysis of the regression models (Equations (3)–(5)) for the microalgae *Scenedesmus* sp. The microalgae were exposed to microplastics (MPs) particles of: polyethylene (PE), polypropylene (PP), polystyrene (PS), polyvinyl chloride (PVC), and polyethylene terephthalate (PET). The analyzed factors were concentration and size of the particles.

MPs	Model					Coefficients						
	No.	*R* ^2^	*R* ^2^ _adj_	*F*	*p*	Coefficient	Value	Lower 95%	Upper 95%	*p*	Significant Term?	Influential Factor
PE	MODEL I	0.9535	0.9458	123.17	0.00	*a* _0_	5.45	4.19	6.70	0.00		concentration & size
						*a*_1_ (10^−3^)	8.26	7.04	9.47	0.00	YES	
						*a*_2_ (10^−3^)	−5.93	−8.46	−3.41	0.00	YES	
	MODEL II	0.9618	0.9465	62.95	0.00	*a* _0_	7.37	4.07	10.67	0.00		concentration
						*a*_1_ (10^−2^)	0.84	0.35	1.33	0.00	YES	
						*a*_2_ (10^−2^)	−1.76	−3.56	0.03	0.94		
						*a*_3_ (10^−6^)	−0.14	−4.64	4.35	0.17		
						*a*_4_ (10^−5^)	1.46	−0.76	3.68	0.05		
	MODEL III	0.9680	0.9502	54.40	0.00	*a* _0_	6.50	2.93	10.07	0.00		concentration
						*a*_1_ (10^−2^)	1.01	0.45	1.57	0.00	YES	
						*a*_2_ (10^−2^)	−1.54	−3.34	0.25	0.08		
						*a*_3_ (10^−6^)	−0.14	−4.55	4.26	0.22		
						*a*_4_ (10^−5^)	1.46	−0.71	3.64	0.94		
						*a*_5_ (10^−5^)	−0.43	−1.17	0.31	0.16		
PP	MODEL I	0.8824	0.8628	45.01	0.00	*a* _0_	3.25	2.02	4.48	0.00		concentration
						*a*_1_ (10^−3^)	5.19	3.99	6.38	0.00	YES	
						*a*_2_ (10^−3^)	−0.48	−2.96	2.00	0.68		
	MODEL II	0.9055	0.8677	23.96	0.00	*a* _0_	3.79	0.58	6.99	0.02		concentration
						*a*_1_ (10^−2^)	0.80	0.33	1.28	0.00	YES	
						*a*_2_ (10^−2^)	−0.63	−2.37	1.11	0.44		
						*a*_3_ (10^−6^)	−2.70	−7.07	1.67	0.20		
						*a*_4_ (10^−5^)	0.72	−1.43	2.88	0.47		
	MODEL III	0.9279	0.8878	23.15	0.00	*a* _0_	2.76	−0.55	6.06	0.09		concentration
						*a*_1_ (10^−2^)	1.00	0.48	1.52	0.00	YES	
						*a*_2_ (10^−2^)	−0.37	−2.04	1.30	0.63		
						*a*_3_ (10^−6^)	−2.70	−6.79	1.39	0.17		
						*a*_4_ (10^−5^)	0.72	−1.29	2.74	0.44		
						*a*_5_ (10^−5^)	−0.51	−1.19	0.18	0.13		
PS	MODEL I	0.8570	0.8332	35.95	0.00	*a* _0_	5.12	2.83	7.41	0.00		concentration & size
						*a*_1_ (10^−2^)	0.80	0.58	1.03	0.00	YES	
						*a*_2_ (10^−2^)	−0.66	−1.12	−0.19	0.01	YES	
	MODEL II	0.9525	0.9335	50.11	0.00	*a* _0_	1.10	0.71	1.48	0.00		concentration & size
						*a*_1_ (10^−2^)	1.20	0.63	1.77	0.00	YES	
						*a*_2_ (10^−2^)	−4.53	−6.62	−2.45	0.00	YES	
						*a*_3_ (10^−6^)	−3.78	−9.01	1.45	0.14		
						*a*_4_ (10^−5^)	4.84	2.26	7.43	0.00	YES	
	MODEL III	0.9565	0.9323	39.54	0.00	*a*_0_ (10)	1.17	0.74	1.60	0.00		concentration & size
						*a*_1_ (10^−2^)	1.06	0.37	1.74	0.01	YES	
						*a*_2_ (10^−2^)	−4.72	−6.90	−2.53	0.00	YES	
						*a*_3_ (10^−6^)	−3.78	−9.14	1.58	0.14		
						*a*_4_ (10^−5^)	4.85	2.20	7.49	0.00	YES	
						*a*_5_ (10^−5^)	0.36	−0.54	1.26	0.39		
PVC	MODEL I	0.9331	0.9219	83.67	0.00	*a* _0_	4.87	3.26	6.48	0.00		concentration & size
						*a*_1_ (10^−2^)	0.86	0.71	1.02	0.00	YES	
						*a*_2_ (10^−2^)	−0.69	−1.02	−0.37	0.00	YES	
	MODEL II	0.9435	0.9208	41.71	0.00	*a* _0_	2.53	−1.76	6.82	0.22		concentration
						*a*_1_ (10^−2^)	1.03	0.40	1.66	0.00	YES	
						*a*_2_ (10^−2^)	0.55	−1.78	2.89	0.61		
						*a*_3_ (10^−6^)	−1.62	−7.47	4.23	0.55		
						*a*_4_ (10^−5^)	−1.56	−4.44	1.32	0.26		
	MODEL III	0.9705	0.9541	59.18	0.00	*a* _0_	0.57	−3.09	4.23	0.73		concentration & size
						*a*_1_ (10^−2^)	1.42	0.84	2.00	0.00	YES	
						*a*_2_ (10^−2^)	1.04	−0.80	2.89	0.23		
						*a*_3_ (10^−6^)	−1.62	−6.14	2.90	0.44		
						*a*_4_ (10^−5^)	−1.56	−3.79	0.67	0.15		
						*a*_5_ (10^−5^)	−0.96	−1.72	−0.20	0.02	YES	
PET	MODEL I	0.8845	0.8653	45.95	0.00	*a* _0_	4.75	3.86	5.63	0.00		concentration & size
						*a*_1_ (10^−3^)	3.17	2.31	4.03	0.00	YES	
						*a*_2_ (10^−3^)	−4.30	−6.09	−2.51	0.00	YES	
	MODEL II	0.9136	0.8790	26.42	0.00	*a* _0_	5.46	3.23	7.69	0.00		concentration
						*a*_1_ (10^−3^)	5.23	1.93	8.53	0.00	YES	
						*a*_2_ (10^−2^)	−1.05	−2.26	0.17	0.08		
						*a*_3_ (10^−6^)	−1.96	−5.01	1.09	0.18		
						*a*_4_ (10^−5^)	0.77	−0.73	2.27	0.28		
	MODEL III	0.9335	0.8965	25.25	0.00	*a* _0_	4.75	2.43	7.06	0.00		concentration
						*a*_1_ (10^−2^)	0.66	0.30	1.03	0.00	YES	
						*a*_2_ (10^−2^)	−0.87	−2.03	0.30	0.13		
						*a*_3_ (10^−6^)	−1.96	−4.82	0.90	0.15		
						*a*_4_ (10^−5^)	0.77	−0.64	2.18	0.25		
						*a*_5_ (10^−6^)	−3.48	−8.27	1.32	0.13		

**Table 3 polymers-14-01246-t003:** Statistical analysis of the regression models (Equations (3)–(5)) for the bacteria *Pseudomonas putida*. The bacteria were exposed to microplastics (MPs) particles of: polyethylene (PE), polypropylene (PP), polystyrene (PS), polyvinyl chloride (PVC), and polyethylene terephthalate (PET). The analyzed factors were concentration and size of the particles.

MPs	Model					Coefficients						
	No.	*R* ^2^	*R* ^2^ _adj_	*F*	*p*	Coefficient	Value	Lower 95%	Upper 95%	*p*	Significant Term?	Influentia Factor
PE	MODEL I	0.8002	0.7669	24.03	0.00	*a* _0_	5.01	2.32	7.69	0.00		concentration & size
						*a*_1_ (10^−2^)	0.75	0.49	1.01	0.00	YES	
						*a*_2_ (10^−2^)	−0.73	−1.27	−0.19	0.01	YES	
	MODEL II	0.8306	0.7628	12.26	0.00	*a*_0_ (10)	0.65	−0.06	1.37	0.07		concentration
						*a*_1_ (10^−2^)	1.25	0.18	2.31	0.03	YES	
						*a*_2_ (10^−2^)	–2.11	–6.01	1.80	0.26		
						*a*_3_ (10^−5^)	−0.47	−1.45	0.51	0.31		
						*a*_4_ (10^−5^)	1.72	−3.11	6.55	0.45		
	MODEL III	0.9228	0.8799	21.52	0.00	*a* _0_	3.04	−2.68	8.76	0.26		concentration & size
						*a*_1_ (10^−2^)	1.93	1.03	2.83	0.00	YES	
						*a*_2_ (10^−2^)	−1.23	−4.11	1.65	0.36		
						*a*_3_ (10^−5^)	−0.47	−1.18	0.23	0.16		
						*a*_4_ (10^−5^)	1.72	−1.77	5.21	0.23		
						*a*_5_ (10^−5^)	−1.72	−2.90	−0.53	0.01	YES	
PP	MODEL I	0.9176	0.9039	66.83	0.00	*a* _0_	4.25	3.28	5.21	0.00		concentration & size
						*a*_1_ (10^−3^)	4.47	3.53	5.41	0.00	YES	
						*a*_2_ (10^−3^)	−4.49	−6.44	−2.54	0.00	YES	
	MODEL II	0.9192	0.8869	28.44	0.00	*a* _0_	3.84	1.06	6.62	0.01		concentration
						*a*_1_ (10^−3^)	5.08	0.98	9.19	0.02	YES	
						*a*_2_ (10^−2^)	−0.26	−1.77	1.25	0.71		
						*a*_3_ (10^−6^)	−0.59	−4.38	3.20	0.74		
						*a*_4_ (10^−5^)	−0.23	−2.10	1.63	0.78		
	MODEL III	0.9379	0.9035	27.20	0.00	*a* _0_	2.95	0.08	5.83	0.04		concentration
						*a*_1_ (10^−2^)	0.68	0.23	1.14	0.01	YES	
						*a*_2_ (10^−2^)	−0.04	−1.49	1.41	0.95		
						*a*_3_ (10^−6^)	−0.59	−4.14	2.97	0.72		
						*a*_4_ (10^−5^)	−0.23	−1.99	1.52	0.77		
						*a*_5_ (10^−5^)	−0.43	−1.03	0.62	0.13		
PS	MODEL I	0.9159	0.9019	65.37	0.00	*a* _0_	6.30	4.22	8.38	0.00		concentration & size
						*a*_1_ (10^−2^)	1.01	0.81	1.21	0.00	YES	
						*a*_2_ (10^−2^)	−0.66	−1.08	−0.24	0.00	YES	
	MODEL II	0.9656	0.9519	70.26	0.00	*a*_0_ (10)	0.68	0.29	1.06	0.00		concentration
						*a*_1_ (10^−2^)	1.91	1.34	2.48	0.00	YES	
						*a*_2_ (10^−2^)	−1.75	−3.85	0.35	0.09		
						*a*_3_ (10^−5^)	−0.86	−1.38	−0.33	0.00	YES	
						*a*_4_ (10^−5^)	1.36	−1.24	3.97	0.27		
	MODEL III	0.9723	0.9570	63.29	0.00	*a* _0_	5.62	1.53	9.72	0.01		concentration
						*a*_1_ (10^−2^)	2.13	1.48	2.77	0.00	YES	
						*a*_2_ (10^−2^)	−1.47	−3.53	0.60	0.14		
						*a*_3_ (10^−5^)	−0.86	−1.36	−0.35	0.00	YES	
						*a*_4_ (10^−5^)	1.36	−1.13	3.86	0.25		
						*a*_5_ (10^−5^)	−0.55	−1.40	0.29	0.17		
PVC	MODEL I	0.8626	0.8397	37.66	0.00	*a* _0_	7.02	5.30	8.75	0.00		concentration & size
						*a*_1_ (10^−3^)	5.07	3.40	6.74	0.00	YES	
						*a*_2_ (10^−2^)	−0.90	−1.24	−0.55	0.00	YES	
	MODEL I	0.9340	0.9076	35.37	0.00	*a*_0_ (10)	1.11	0.76	1.46	0.00		concentration & size
						*a*_1_ (10^−2^)	0.72	0.21	1.24	0.01	YES	
						*a*_2_ (10^−2^)	−3.53	−5.42	−1.64	0.00	YES	
						*a*_3_ (10^−6^)	−2.05	−6.78	2.68	0.36		
						*a*_4_ (10^−5^)	3.29	0.96	5.63	0.01	YES	
	MODEL III	0.9589	0.9360	41.97	0.00	*a*_0_ (10)	0.97	0.64	1.29	0.00		concentration & size
						*a*_1_ (10^−2^)	1.00	0.49	1.51	0.00	YES	
						*a*_2_ (10^−2^)	−3.18	−4.81	−1.55	0.00	YES	
						*a*_3_ (10^−6^)	−2.05	−6.05	1.95	0.28		
						*a*_4_ (10^−5^)	3.29	1.32	5.27	0.00	YES	
						*a*_5_ (10^−5^)	−0.69	−1.36	−0.02	0.04	YES	
PET	MODEL I	0.8016	0.7686	24.25	0.00	*a* _0_	6.00	4.24	7.75	0.00		concentration & size
						*a*_1_ (10^−3^)	3.13	1.43	4.84	0.00	YES	
						*a*_2_ (10^−2^)	−0.92	−1.28	−0.57	0.00	YES	
	MODEL II	0.8850	0.8389	19.23	0.00	*a* _0_	1.77	−2.12	5.66	0.33		concentration & size
						*a*_1_ (10^−2^)	0.61	0.04	1.19	0.04	YES	
						*a*_2_ (10^−2^)	1.34	–0.77	3.46	0.19		
						*a*_3_ (10^−6^)	−2.83	−8.14	2.47	0.26		
						*a*_4_ (10^−5^)	−2.83	−5.45	−0.22	0.04	YES	
	MODEL III	0.9246	0.8827	22.07	0.00	*a* _0_	0.26	-3.46	3.97	0.88		concentration & size
						*a*_1_ (10^−2^)	0.91	0.32	1.49	0.01	YES	
						*a*_2_ (10^−2^)	1.72	−0.15	3.59	0.07		
						*a*_3_ (10^−6^)	−2.83	−7.43	1.76	0.20		
						*a*_4_ (10^−5^)	−2.83	−5.10	−0.57	0.02	YES	
						*a*_5_ (10^−5^)	−0.74	−1.51	0.03	0.06		

**Table 4 polymers-14-01246-t004:** Statistical analysis of the regression models (Equations (3)–(5)) for the yeast *Saccharomyces cerevisiae*. The yeast was exposed to microplastics (MPs) particles of: polyethylene (PE), polypropylene (PP), polystyrene (PS), polyvinyl chloride (PVC), and polyethylene terephthalate (PET). The analyzed factors were concentration and size of the particles.

MPs	Model					Coefficients						
	No.	*R* ^2^	*R* ^2^ _adj_	*F*	*p*	Coefficient	Value	Lower 95%	Upper 95%	*p*	Significant Term?	Influential Factor
PE	MODEL I	0.9449	0.9358	102.97	0.00	*a*_0_ (10)	6.97	5.83	8.10	0.00		concentration & size
						*a*_1_ (10^−2^)	5.78	4.68	6.87	0.00	YES	
						*a*_2_ (10^−1^)	−0.90	−1.13	−0.67	0.00	YES	
	MODEL II	0.9804	0.9725	124.86	0.00	*a*_0_ (10)	3.90	1.93	5.86	0.00		concentration & size
						*a*_1_ (10^−1^)	0.93	0.64	1.22	0.00	YES	
						*a*_2_ (10^−1^)	0.62	−0.45	1.69	0.22		
						*a*_3_ (10^−5^)	−3.35	−6.03	−0.67	0.02	YES	
						*a*_4_ (10^−4^)	−1.90	−3.23	−0.58	0.01	YES	
	MODEL III	0.9860	0.9782	126.77	0.00	*a*_0_ (10)	4.59	2.63	6.55	0.00		concentration & size
						*a*_1_ (10^−1^)	0.79	0.48	1.10	0.00	YES	
						*a*_2_ (10^−1^)	0.45	−0.54	1.43	0.33		
						*a*_3_ (10^−5^)	−3.35	−5,77	−0.93	0.01	YES	
						*a*_4_ (10^−4^)	−1.90	−3.10	−0.71	0.01	YES	
						*a*_5_ (10^−5^)	3.42	−0.65	7.48	0.09		
PP	MODEL I	0.8100	0.7784	25.58	0.00	*a*_0_ (10)	5.57	3.87	7.27	0.00		concentration & size
						*a*_1_ (10^−2^)	4.72	3.07	6.37	0.00	YES	
						*a*_2_ (10^−2^)	−5.50	−8.93	−2.07	0.00	YES	
	MODEL II	0.9387	0.9142	28.28	0.00	*a*_0_ (10)	2.65	−0.16	5.46	0.06		concentration
						*a*_1_ (10^−1^)	1.25	0.83	1.67	0.00	YES	
						*a*_2_ (10^−1^)	0.49	−1.03	2.02	0.49		
						*a*_3_ (10^−4^)	−0.74	−1.13	−0.36	0.00	YES	
						*a*_4_ (10^−4^)	−1.30	−3.19	0.59	0.15		
	MODEL III	0.9580	0.9347	41.05	0.00	*a*_0_ (10)	3.69	0.95	6.44	0.01		concentration
						*a*_1_ (10^−1^)	1.05	0.61	1.48	0.00	YES	
						*a*_2_ (10^−1^)	0.23	−1.15	1.62	0.71		
						*a*_3_ (10^−4^)	−0.74	−1.08	−0.40	0.00	YES	
						*a*_4_ (10^−4^)	−1.30	−2.98	0.37	0.11		
						*a*_5_ (10^−4^)	0.51	−0.06	1.08	0.07		
PS	MODEL I	0.9334	0.9223	84.09	0.00	*a*_0_ (10)	5.88	4.65	7.11	0.00		concentration & size
						*a*_1_ (10^−2^)	6.32	5.13	7.52	0.00	YES	
						*a*_2_ (10^−2^)	−6.79	−9.27	−0.04	0.00	YES	
	MODEL II	0.9468	0.9255	44.50	0.00	*a*_0_ (10)	3.86	0.65	7.06	0.02		concentration
						*a*_1_ (10^−1^)	0.79	0.32	1.27	0.00	YES	
						*a*_2_ (10^−1^)	0.39	−1.36	2.13	0.63		
						*a*_3_ (10^−5^)	−1.55	−5.92	2.82	0.45		
						*a*_4_ (10^−4^)	−1.33	−3.49	0.82	0.20		
	MODEL III	0.9687	0.9514	55.76	0.00	*a*_0_ (10)	5.22	2.32	8.12	0.00		concentration & size
						*a*_1_ (10^−2^)	5.28	0.71	9.85	0.03	YES	
						*a*_2_ (10^−1^)	0.05	−1.42	1.51	0.94		
						*a*_3_ (10^−5^)	−1.55	−5.14	2.04	0.35		
						*a*_4_ (10^−4^)	−1.33	−3.10	0.44	0.12		
						*a*_5_ (10^−4^)	0.67	0.06	1.27	0.03	YES	
PVC	MODEL I	0.8314	0.8033	29.59	0.00	*a*_0_ (10)	6.61	4.71	8.52	0.00		concentration & size
						*a*_1_ (10^−2^)	4.74	2.89	6.58	0.00	YES	
						*a*_2_ (10^−1^)	−0.93	−1.32	−0.55	0.00	YES	
	MODEL II	0.9672	0.9541	73.75	0.00	*a*_0_ (10^2^)	1.29	1.05	1.54	0.00		concentration & size
						*a*_1_ (10^−2^)	0.10	−3.51	3.71	0.95		
						*a*_2_ (10^−1^)	−4.31	−5.64	−2.98	0.00	YES	
						*a*_3_ (10^−5^)	4.42	1.09	7.75	0.01	YES	
						*a*_4_ (10^−4^)	4.22	2.57	5.86	0.00	YES	
	MODEL III	0.9746	0.9605	69.01	0.00	*a*_0_ (10^2^)	1.37	1.12	1.63	0.00		concentration & size
						*a*_1_ (10^−2^)	−1.40	−5.41	2.60	0.45		
						*a*_2_ (10^−1^)	−4.50	−5.78	−3.22	0.00	YES	
						*a*_3_ (10^−5^)	4.42	1.28	7.56	0.01	YES	
						*a*_4_ (10^−4^)	4.22	2.67	5.77	0.00	YES	
						*a*_5_ (10^−5^)	3.76	−1.51	9.03	0.14		
PET	MODEL I	0.9596	0.9528	142.33	0.00	*a*_0_ (10)	5.61	4.71	6.52	0.00		concentration & size
						*a*_1_ (∙10^−2^)	5.54	4.66	6.42	0.00	YES	
						*a*_2_ (∙10^−1^)	−0.82	−1.00	−0.64	0.00	YES	
	MODEL II	0.9889	0.9844	222.23	0.00	*a*_0_ (10)	3.98	2.59	5.36	0.00		concentration
						*a*_1_ (10^−1^)	0.98	0.78	1.19	0.00	YES	
						*a*_2_ (10^−2^)	−2.29	−9.81	5.23	0.51		
						*a*_3_ (10^−5^)	−4.08	−5.96	−2.19	0.00	YES	
						*a*_4_ (10^−4^)	−0.74	−1.67	0.19	0.11		
	MODEL III	0.9944	0.9913	318.85	0.00	*a*_0_ (10)	3.33	2.17	4.49	0.00		concentration & size
						*a*_1_ (10^−1^)	1.11	0.93	1.29	0.00	YES	
						*a*_2_ (10^−2^)	−0.68	−6.53	5.17	0.80		
						*a*_3_ (10^−5^)	−4.08	−5.51	−2.64	0.00	YES	
						*a*_4_ (10^−4^)	−0.74	−1.45	−0.03	0.04	YES	
						*a*_5_ (10^−5^)	−3.16	−5.56	−0.75	0.02	YES	

## Data Availability

Data are contained within the article.

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
