# Peer review of "Assessment of the Influence of Size and Concentration on the Ecotoxicity of Microplastics to Microalgae Scenedesmus sp., Bacterium Pseudomonas putida and Yeast Saccharomyces cerevisiae"

_polymers, 2022, doi:10.3390/polym14061246_

Round 1

Reviewer 1 Report

I have completed the review of this article. The authors used the model to evaluate the three microbial growth processes under the influence of microplastics. The research methods and conclusions are not problematic, but there are little details should be addressed. This study requires minor revisions for publication on this journal.

  1. The title is inappropriate, and the specific species name is a better choice.
  2. There are problems with choosing the keywords and I suggest replacing them.
  3. The authors should briefly introduce the exposure and threat of microplastics, please consider the following last references:

10.1016/j.jhazmat.2021.126843

10.1016/j.scitotenv.2021.148200

10.1016/j.marpolbul.2021.112738

10.1016/j.scitotenv.2019.02.132

  1. Figure 1 and 2, these two figures should be moved to Supplementary.
  2. Conclusion should be briefer and more precise.

Author Response

We would like to thank the Reviewer no. 1 for his valuable comments. We accepted all the suggestions and corrected the manuscript accordingly.

Reviewer 2 Report

Dear Authors, the research related to the ecotoxicity of microplastics is of high significance knowing that plastic pollution is a serious globally recognized environmental problem. In my opinion, the research reflection in the manuscript is quite well organized, only minor corrections need to be performed. The abstract should avoid needless verbiage and needs targeted focus on the research, e.g., the second sentence (line 11-12) is unnecessary.  In the abstract, it is better also to indicate abbreviations of plastics in brackets, the same as in lines 35-37. The manuscript should be checked for correct reference reflection in the text, e.g., in lines 38, 42, 45 and other places, punctuation should be corrected to [x] followed by a dot. Lines 73-84 as well as Figures 1 and 2 should be deleted from the introduction and included in the materials and methods. Also lines 501-507 better fit into the methodology and are not necessary for the conclusions.

Author Response

We would like to thank the Reviewer no. 2 for his valuable comments. We accepted all the suggestions and corrected the manuscript accordingly.

Reviewer 3 Report

The manuscript by M. Miloloža et al., 'Ecotoxicological assessment of microplastics influence on three microorganisms: microalgae, bacterium and yeast' attempts to look at the toxicity effects of different polymers on three different organisms. While such information would be valuable to the research community and general public, the experiments were designed such that these questions are not actually addressed. In short, they lacked non-plastic controls, did not confirm the polymer type of their plastics and also, amazingly, did not sterilize the plastic samples prior to application to test organisms. There were also major issues with the format of the manuscript, the figures/tables, the discussion, the use of citations and the English. Details about all these issues are included below. 

Major Issues:

1. Lack of proper controls. 

The point of this paper was to test the ability of 5 plastic polymer types (PE, PP, PS, PVC and PET) to inhibit the growth of three organisms- Scenedesmus sp., Pseudomonas putida, and Saccharomyces cerevisiae. These 5 polymers were fractionated into three size levels (200, 400, and 600 μm) and applied to the organisms at 5 concentration levels (50, 250, 500, 750, and 98 1000 mg/L). Some combinations of these polymers, size and concentrations lead to inhibition of growth and some did not. But does this mean that it is the plastics themselves that are leading to the inhibition of growth? The paper says yes, however, it is impossible to determine this based on these experiments. That is due to the complete lack of a non-plastic controls.

If you were to coat any microorganism with a high concentration of very small beads, you are likely to see some inhibition of growth. Since this manuscript includes no non-plastic samples, say like glass, wood, chitin, etc., then all this paper shows is that being grown surrounded by small hard objects impedes growth. How do the authors know that there is no shearing occurring during the shaking of the flasks? 160 rpm is fairly rapid and just the act of shaking a flask full of bacteria coated in hard objects might lead to cellular breakage. 

2. Growth kinetics 

Why were the three organisms studied using the exact same time span? They in no way have the same growth kinetics. For example, P. putida will have likely gone through exponential growth, stationary phase and death phase by the first 24 hours alone. I looked up the guidelines for the P. putida Growth Inhibition Test and I believe that growth is supposedly to be evaluated after 16 hours, not 72 hours like it is in this manuscript. 

3. The plastic samples

How did the author's obtain these plastic pieces? Are they from the store, straight from the manufacturer or were they environmentally sampled? How do they know what they are exactly made of? Did they test them? Manufacturers sometimes are not 100% aware or honest about the exact polymer composition of their plastic items. I have personally experienced manufacturers stating that an item is composed of one type of polymer, and when we measured it by AFT-FTIR and RAMAN, we discovered they were another polymer entirely. 

However the author's obtained the plastic pieces, they make make no mention of how they sterilized the plastic samples before they are applied to the three test organisms. Hopefully this detail was just omitted in the methods, but, if not, then there can be no way to know whether the inhibition they are seeing is due to the objects OR to the microorganisms that were already on the objects. Thus the inhibition in growth in the test organisms could be either due to infection or competition for nutrients.

And if the applied plastic were not sterilized, how do the author's know that the colonies that grew up from these flasks three days after growth are still all P. putida? There was no mention of how they determined whether the organisms at the end of the experiment matched the ones at the beginning. Even if the plastic pieces were sterilized, contaminations happen. 

4. Discussions about biodegrading plastic 

Throughout the paper, the author's speculate widely, and inappropriately, about the test's organisms ability to biodegrade plastic within the confines of the experiments. Each time a stated polymer type did not inhibit a test organism, the authors suggest this could be due to them consuming the plastic as a nutrient source. They even cite published literature to back these claims. However, if they had read the literature they are citing, they would know how impossible it would be for any of these organisms to consume the polymers within these experiments. 

For example, they cite Giacomucci, et. al., 2019 paper: 'Polyvinyl chloride biodegradation by Pseudomonas citronellolis and 639 Bacillus flexus' as evidence that it is possible that the reason why PVC did not inhibit the growth P. putida within this study was due to P. putida biodegrading PVC. However, the experiments within Giacomucci et al., were conducted over 45 to 90 days. The experiments in this manuscript were conducted over 3 days. Furthermore, the experiments in Giacomucci, 2019 used an entirely different species of Pseudomonas. The Pseudomonas genus is one of the most widely genetically diverse genus of Gram-negative bacteria. Just because one species has metabolic capabilities, does not mean it make sense to apply these same metabolic capabilities to a completely different species. 

In general, if any form of biodegradation and / or depolymerization was possible within 3 days in uncontrolled conditions, then there would be no problem with plastic pollution build up in the environment

5. Structure of manuscript

After reading this manuscript I was generally confused of its main purpose. Based on the abstract and introduction, the goal of the manuscript would seem to test the toxicity of five polymers on three organisms. However, the majority of the results and discussion are spent on the validity of three different models of measuring growth inhibition. There is so much discussion on the different models that it is very hard to figure out what the actual results are. 

If the point of the paper is to valid a model, then this should be stated up front. But if it is to test toxicity, then there needs to be WAY less discussion about the models and just clear cut results about the toxicity. 

To this point, if this paper is about toxicity, then the information in the tables should be supplementary, at best, and there should be a single clear table revealing the results. 

Also, the figures are hard to read. Sometimes actual data points are not visible. See Figure 4E for an example. There is no need for these to be three dimensional. Just simple 2D graphs are fine. 

6. Use of citations

I have already provided examples of where citations were used inappropriately, such as to defend their experimental design and speculative conclusions. However, I also found a case where the authors cited themselves inappropriately. The inclusion of reference 40 on lines 412-414 does not make sense. The line states ' Pseudomonas putida are able to metabolize ethlyene glycol [39] which is one of the products of PET depolymerization [40].' I was super curious under what circumstances of depolymerization they were discussing, so I looked up reference 40. It is a review with the same first author of this manuscript. And I could not find a single reference to depolymerization or ethylene glycol throughout the review. 

7. English 

This paper had numerous typos throughout. Articles like 'the' and 'an' were often missing. Words were often left singular when they should be plural and vice versa. The language often leaned towards being informal and sometimes the word choice was questionable (ie. the continued use of the word 'adversity'). I tried to list some of the mistakes I found, but I am sure I didn't catch them all. Though I am a native English speaker, it is not the reviewer's job to proofread and correct all the grammatical mistakes in a manuscript. Below are some line to line corrections. I am sure I did not catch them all. 

Line by line corrections: 

-throughout abstract: The species names of the organisms should be italicized. 

-line 72: 'of' is missing

-line 78: Why is microorganisms bolded?

-line 193: the reference should be within the period 

-line 202: 'weather' should be 'whether' 

-line 205: 'Table' is repeated 

-line 211: 'figure' is repeated again; or maybe parentheses are missing, its not clear 

-line 289: add 'an' before 'experiment'

-line 303: add 'the' in front of 'case'

-line 304: add 'to' in front of 'concentration'

-line: 309: 'In opposite' is not a phrase commonly used in English. Perhaps, 'In contrast'? 

-lines 325-326: This seems highly speculative. Over what sort of time period does PET hydrolysis occur? I find it highly unlikely that PET hydrolysis would begin after 3 days of growth. 

-lines 357-359: Once again, this discussion about biodegradation is highly speculative and doesn't make sense within a manuscript about toxicity. 

-lines 375: remove 'practically'; informal 

-lines 392: replace 'generally' with 'the'

-line 403-405: Once again, this discussion of biodegradation is wholly unfounded within the confines of these experiments. Giacomucci found biodegradation after 45 to 90 days. These experiments were done over 3 days. 

- 412 - 418: Everything within these lines is purely speculative. In fact, depolymerization of PET within the confines of these experiments is likely impossible. And the author's misuse the reference 40.

- 432-433: 'Was it true, or not,...' informal and also grammatically incorrect 

- 491: 'This practically concerned' poorly worded 

-lines 528 - 529: 'the one with'; unscientific phrasing, how about 'the model with'

-line 531: remove the 'thus'; 'scientist' should be 'scientists'

-lines 530 - 533: This sentence is poorly worded and contains multiple typos.

Author Response

We would like to thank the Reviewer no. 3 for his valuable comments. We accepted all the suggestions and corrected the manuscript accordingly.

Round 2

Reviewer 3 Report

Comment 1: Lack of proper controls.

Answer:

The Reviewer obviously overlooked the use of non-plastic controls because they are indicated for each of the toxicity tests.

  • For Scenedesmus sp. (line 152 of the old version of the manuscript): “The test required control flask, which was used for comparison. Control flask did not contain MPs particles, but the suspension of algae and basal medium only.

  • For Pseudomonas putida (line 161 of the old version of the manuscript): “Analogous to the previously described algal test, the control flasks were prepared as well.

  • For Saccharomyces cerevisiae (line 172 of the old version of the manuscript): “Control bottles were used as well.”

In addition, the Reviewer believes that 160 rpm is fast and that shaking a flask full of bacteria coated in hard objects might lead to cellular breakage. Rodriguez et al. (Rodriguez, A.; Escobar, S.; Gomez, E.; Santos, V.E.; Garcia-Ochoa, F. Behavior of several pseudomonas putida strains growth under different agitation and oxygen supply conditions. Biotechnol. Prog. 2018, 34, 900–909.) studied the effect of agitation on bacterial growth; the microorganism used was Pseudomonas putida. The study was conducted in the range of 100-800 rpm. Rodriguez reported that increasing the agitation enhanced the growth rate of Pseudomonas within the applied speed range, as the amount of oxygen played an important role in cell growth.

Reviewer Response:

The author’s did not address my issues under this point at all. I did not overlook the inclusion of control flasks that contain only the organism and base medium. The lack of controls I was referring to were the lack of non-plastic particle controls.

To quote myself from my first review, ‘If you were to coat any microorganism with a high concentration of very small beads, you are likely to see some inhibition of growth.’ To control for particle effects, the author’s should have included control flasks that contained non-plastic controls, such as (to quote myself again), ‘glass, wood, chitin, etc.,’ Other possibilities would have been stone or silica. These non-plastic particles should have been in the same concentration and size distribution as the polymer types. The author’s state that they are measuring the ‘ecotoxicity of microplastics’, specifically, the toxicity of PE (C₂H₄)ₙ, PP (C3H6)n, PS (C8H8)n, PVC (C2H3Cl)n, PET (C10H8O4)n. However, because the authors do not include any particles that are not composed of plastic polymers, all these experiments show is that if you grown Pseudomonas putida, Saccharomyces cerevisiae and Scenedesmus sp. in the presence of high concentrations of small beads, this might change their growth patterns. I say 'might' here because the author's designed the experiments so that Pseudomonas putida and Saccharomyces cerevisiae were in the death phases of their growth, and thus, they really cannot comment on how these polymer changed their growth rates.

The authors also completely did not understand or purposefully did not respond to my point that if you shake a flask of bacteria that is covered in microbeads, chances are there is going to be some physical shearing. They cite a paper where Pseudomonas sp.are not being grown in suspension of small beads as a response to the point I made. This does not make any sense.

When extracting DNA from microorganisms, we use a technique called bead beading where you shear the cellular membrane to release the DNA by shaking bacteria with beads very quickly. I am not saying they are doing exactly this in the paper, but how do they not control for some level of mechanical breakage by the cells in response to the small particles? This would lead to an inhibition of growth by something other than the direct effect of the polymer. They could have controlled for this, partly, by including non-plastic particle controls. If they had seen the same level of growth inhibition by growing the organisms in a suspension of small glass beads, then they would know it was a particle effect, not a polymer effect.

In summary, the authors still do not have the proper controls.

Comment 2: Growth kinetics

Why were the three organisms studied using the exact same time span? They in no way have the same growth kinetics. For example, P. putida will have likely gone through exponential growth, stationary phase and death phase by the first 24 hours alone. I looked up the guidelines for the P. putida Growth Inhibition Test and I believe that growth is supposedly to be evaluated after 16 hours, not 72 hours like it is in this manuscript.

Answer:

We would like to thank the Reviewer for this comment, as he very well noticed that additional explanation on the performed toxicity test with Pseudomonas putida is missing. The toxicity test with Pseudomonas putida originally refers to solutions and not suspensions as it was the case in our study. Therefore, we performed some preliminary experiments with Pseudomonas putida to see if the standard method was applicable. The preliminary experiments showed that CFU did not change within 16 hours, which was not the case after 72 hours. The same situation occurred with theSaccharomyces test. In addition, the 72-hour exposure allowed us to more easily compare the inhibitions of all three microorganisms used, since they all had the same exposure time.

Based on the Reviewer’s comment, we added additional sentences to the methodology (chapter 2.1. Design of the experiment; lines 123-131 of the corrected manuscript):

The standard toxicity test with Scenedesmus sp. lasts 72 hours, whereas the duration of the standard tests with Pseudomonas putida and Saccharomyces cerevisiae is 16 hours. However, the tests with Pseudomonas putida and Saccharomyces cerevisiae originally refer to solutions and not suspensions, as was the case in our study. Therefore, we performed some preliminary experiments for these two tests to see if the standard methods were applicable. The preliminary experiments showed that CFU did not change within 16 hours, which was not the case at 72 hours. Therefore, we set an identical contact time of 72 hours for all three tests. An identical contact time also allowed us to more easily compare inhibitions between the three microorganisms used.

Reviewer Response

The author's response to this also makes no biological sense and just leads to so many more questions. I should state that I worked with Pseudomonas sp., including P. putida, for years. While working with these bacteria I often undertook the process of creating very detailed growth curves of individual strains by hand, so I have quite a bit of knowledge about the growth kinetics of these bacteria. Are the authors saying they created growth curves of P. putida under each of the experimental tests and there were no growth after 16 hours? And then growth started after 16 hours and continued for 72 hours? I would really like to see the growth curves showing this. Which part of the growth process was so extended? Did the lag phase extend for 48 hours, or did the exponential phase continue for 48 hours? Also, how is the control flask working in this case? The bacterial culture without the polymer suspension would have reached death phase by no longer than 18 hours; so how can comparisons be made between the control flask and the experimental flasks at 72 hours?

If the plastic polymers really did disrupt the traditional growth kinetics of these organisms so drastically, then I recommend showing these growth curves in the results. It would have been much more simple and convincing way to show how the polymer inhibit (or not inhibit) growth. Non-plastic particle controls included, of course.

Also, I am very surprised the authors would include a sentence like this: "An identical contact time also allowed us to more easily compare inhibitions between the three microorganisms used.

The comparison shouldn't be between the organisms at a set time, but at comparable places in their growth phases, say when all organisms are in late exponential phase. This will be at a different time point for each organism.

Comment 3: The plastic samples

Answer:

  • Indeed, we forgot to write some information about characterization of the plastics. The plastics were from the store and the characterization was done by FTIR analysis. We added this information to the manuscript.

    • Lines 135-136 of the corrected manuscript: “These products were purchased from the store.”

    • Lines 140-143 of the corrected manuscript: “The Attenuated Total Reflectance Fourier Transform Infrared (ATR-FTIR) spectroscopic analysis (Spectrum One, Perkin Elmer) was performed to verify the type of the plastics. The characteristic ATR-FTIR spectra are shown in Figure S2.”

Reviewer Response: Thank you for including this information. When doing plastic research, including the exact source of the plastics and validating their composition is very important. The author's should also include the exact type and brand of the plastic that werepurchased. The point of the methods is so that another researcher can mimic the experiments precisely.

Unfortunately, the author's response opens up a new issue.

If If the author's wanted to test the toxicity of the polymers- PE (C₂H₄)ₙ, PP (C3H6)n, PS (C8H8)n, PVC (C2H3Cl)n, PET (C10H8O4)n- they should have acquired these directly from the manufacturer. These are called naive or virgin plastics, and are as close to the pure version of these polymers as is readily available. Commercially available plastics (like those from the store) often contains additives, pigments, and additional chemical compounds. All plastic, naive and commercially available, also contain monomers from the imperfect polymerization process. In addition, plastics are known to absorb chemicals from the surrounding environments and leach them out of their surface when put in liquid. That is why when careful plastic research is conducted, the % of additives / plasticizers are included. For example, in Giacomucci, 2019, a manuscript the author's cite to justify their biodegradation claims, this is how they describe their plastic samples:

"Four petroleum-derived plastic films were used, namely PE, PP, PS and PVC. PVC film contained about 30%w/w of additives/plasticizers, PS contained < 0.5%w/w of mineral oil, high density PE (HDPE) about 0.5%w/w of 1-hexene, linear low density PE (LLDPE) about 4%w/w of butane, while low density PE (LDPE) and PP did not contain any additive components."

As you can see from this, additives and plasticizers can be upwards of 30% of the weight of the plastic and thus should be reported.

Knowing this, how do the author's know that the inhibition of growth they are seeing is due to the polymer and not the leaching of monomer, dyes and additives from the plastic into the medium? By using commercially available plastics they are unable to conclusively say that the affects shown in the manuscript are due to the interactions of the bacteria, fungus and microalgae with the polymers.

Accordingly, we added Figure S2 to Supplementary Materials.

  • The manuscript already contains information how the plastic particles were sterilized (lines 146-150 of the corrected manuscript):

The flasks were filled with 70 % ethanol and shaken on a rotary shaker (Unimax 1010, Heidolph, Germany) at 160 rpm for 10 min to sterilize the MPs particles. The sterilized particles were separated from the ethanol suspension by vacuum membrane filtration using cellulose nitrate 0.45 µm sterile filters (ReliaDiscTM, Ahlstrom-Munksjö, Finland) and washed with sterile deionized water.”

The Reviewer probably overlooked this.

Reviewer Response: The author's are correct, I missed this part of the part of the methods. However, in the future, I would highly recommend double checking the sterilization process by simply plating some of the plastics after the sterilization process Especially since it is well known that spores are not always sterilized by ethanol alone.

Comment 4: Discussions about biodegrading plastic

Answer:

The Reviewer thought some of our suggestions were too speculative. We appreciate the Reviewer’s comment and have changed the text to be less speculative. However, some points need to be discussed, as there is obviously a misunderstanding.

First, we would like to point out that we have never claimed that complete biodegradation occurs within 3 days, nor have we performed such an experiment. However, we cannot exclude the possibility that polymer degradation began during this time period. We are aware that a period of 3 days may seem too short for the onset of hydrolysis, but we found supporting reports in the literature.

Namely, the Reviewer claimed that we supported our suggestion about the potential biodegradation of PVC by the report of Giacomucci et al. (2019), in which experiments were conducted over a much longer period than in our case. In fact, the mentioned period of 45-90 days referred to the preliminary experiment and not to the main experiment. The main experiment lasted 30 days, and during this time a partial PVC biodegradation of 13.07-18.58% was observed. Giacomucci et al. did not mention the time of onset of biodegradation, but they reported the CFU values of the samples and the control. The CFU value of the control started to decrease after the 3rd day of exposure, while the CFU value of the sample remained the same as at the beginning of the experiment. This confirms that biodegradation of PVC started very early in Giacomucci’s study. However, based on the Reviewer’s comment, we realized that the discussion in our manuscript was not clear enough and probably confusing, so we rewrote this part of the manuscript. The new text reads (lines 439-445 of the corrected manuscript):

Giacomucci et al. [47] reported that the bacteria Pseudomonas citronellolis are able to degrade PVC, and based on the reported CFU values, it is clear that biodegradation started very early, practically at the 3rd day of exposure. Therefore, it cannot be excluded that the slightly lower inhibition levels of Pseudomonas putida that we observed during PVC exposure were due to the onset of PVC biodegradation. However, this remains to be confirmed by future studies.”

Reviewer Response:

The authors are completely misinterpreting and misrepresenting the results in Giacomucci et al., 2019. This is a direct quote from the conclusions of Giacomucci et al.: 'Results showed that the biodegradation activity was mainly directed towards PVC additives and to a lesser extent against the PVC polymer chains, as shown by the decrease in Mn.' Giacomucci et al clearly shows that a reduction in some of the PVC polymer chains did occur, but the growth of the bacteria was not shown to be a direct results of the polymer biodegradation and they could not rule out that the bacteria were breaking down the much more readily available additives, plasticizes and readily occurring monomers.

The author's statement: “Giacomucci et al. [47] reported that the bacteria Pseudomonas citronellolis are able to degrade PVC, and based on the reported CFU values, it is clear that biodegradation started very early, practically at the 3rd day of exposure.' isnot based on the results presented in Giacomucci et al., 2019. I have read this manuscript in full and Giacomucci et al., are very careful to make sure that they do not say that biodegradation is happening in full at ANY point and they are very careful to say that they can only measure partial reduction in the polymer chain ('from 100% to 90.87% ± 4.54%) after 90 days. Just because the CFUs did not change does not mean that the bacteria were capable of using PVC as a carbon source!!!!!

Finally it is very important to point out that PVC, or any polymer degradation, would NEVER happen in the experiments presented in Miloloža et al. The author's supplied the bacteria with mineral media. Though the authors do not state what this media contains (a large oversight by the authors), most mineral media either contain yeast or glucose, which are the very readily available forms of carbon. Plastic polymers, in contrast are extremely chemically stable and very difficult to break down. It is impossible for a bacteria to break a plastic polymer down, even partially, in three days, when there is a more readily available carbon source. 

The similar is for the case of Scenedesmus exposed to PET. We found a report by Moog et al. (Moog, D.; Schmitt, J.; Senger, J.; Zarzycki, J.; Rexer, K.-H.; Linne, U.; Erb, T.; Maier, U.G. Using a marine microalgae as a chassis for polyethylene terephthalate (PET) degradation. Microb. Cell. Fact. 2019, 18, 171). Moog exposed the photosynthetic microalgae Phaeodactylum tricornutum to PET and observed a progressive increase in the concentrations of mono(2-hydroxyethyl) terephthalic acid and terephthalic acid, which are the main products of the PET hydrolysis, after only 3 days of exposure. Apparently, some photosynthetic microalgae are able to adapt to PET within such a short period of time. We are aware that Moog’s study was not performed on Scenedesmus and that our statement about the potential star of biodegradation remains only an idea that may be interesting for further studies. Therefore, we have tried to be less speculative and have supplemented the related discussion with facts from the Moog’s report. The related discussion now reads (lines 357-365 of the corrected manuscript):

Recent researches [43,44] have confirmed that some microalgae can produce PET hydrolyzing enzymes called PETases and use PET as substrate. Although it is difficult to claim that this was the case in our experiment without performing a detailed analysis, especially when dealing with a 3-day exposure period, this assumption cannot be dismissed. For example, Moog et al. [43] exposed the photosynthetic microalgae Phaeodactylum tricornutum to PET and observed a progressive increase in the concentrations of mono(2-hydroxyethyl) terephthalic acid and terephthalic acid, which are the main products of the PET hydrolysis, after only 3 days of exposure.

Reviewer Response:

Did the author's actually read Moog et al., 2019.? Moog et al used a genetically engineered microalgae. Moog et al., 2019 took a PETase from a bacteria and transformed Phaeodactylum tricornutum using a plasmid. It is not a naturally occurring PETase from a microalgae. In fact, it is from the bacterium Ideonella sakaiensis, the only microorganism EVER found to have the entire enzymatic pathway to break down a plastic polymer and utilize it as the sole carbon source. I recommend the authors read'Yoshida S, Hiraga K, Takehana T, Taniguchi I, Yamaji H, Maeda Y, Toyo hara K, Miyamoto K, Kimura Y, Oda K. A bacterium that degrades and assimilates poly(ethylene terephthalate). Science. 2016;351:1196–9.'

The other manuscript the authors cite for biodegradation by microalgae (citation #44) by Falah et al., 2020, is also inappropriate, because the plastics were pre-treated prior to being exposed to microalgae. Furthermore, true biodegradation was not measured in Falah et al., 2020.

In conclusion, the authors can cite no corresponding literature to suggest that biodegradation is occurring in these studies. And based on what is currently known about plastic degradation it is impossible that it is occurring in any way in these experiments.

Instead of trying to use biodegradation to explain inconclusive results, the authors should maybe look at the design of their experiments. They did not control for particle effects. They did not control for additives, plasticizers, etc., leaching from the store-bought plastics. They grew all three, very genetically diverse, organisms under the exact same growth time span, even though biologically they do not exhibit the same growth kinetics. All of these could have lead to the very inconsistent results.

Finally, and I can't believe I have to point this out, but it is ridiculous to both show that a chemical has an inhibition effect on growth (PVC in Table 1) and also claim it is a nutrient source for growth. It can't be both.

Comment 5: Structure of manuscript

Answer:

We would like to thank the Reviewer for this valuable comment.

Indeed, our intention was to experimentally determine the inhibitions at different levels of MPs sizes and concentrations, to subsequently use these data for modeling of the inhibition response surface, and finally to determine the significance of the influence of MPs size and concentration within the experimental range. Therefore, based on the Reviewer’s suggestion, we decided to state this in the title of the manuscript. The new title is:

Assessment of the influence of size and concentration on the ecotoxicity of microplastics to microalgae Scenedesmus sp., bacterium Pseudomonas putida and yeast Saccharomyces cerevisiae”.

3D figures from the manuscript show the inhibition response surfaces for experiments in which the influence of two parameters was studied. However, in the originally submitted version of the manuscript, the authors decided to supplement the response surfaces with experimentally obtained inhibition data. The intention was to provide better insight into the goodness-of-fit of the mathematical models applied. In fact, it was not necessary to add the experimental data to the 3D figures because the adjusted R2values listed in Tables 1-3 were sufficient to confirm the goodness-of-fit. Moreover, these experimental data ware already given as supplementary material. It seems that the decision to compile the experimentally determined inhibitions with the response surfaces was not the best one, because the experimental data were sometimes below the response surface and therefore hardly visible, which obviously confused the readers (as the Reviewer commented on Figure 4E). Therefore, we decided to submit the 3D figures containing the response surfaces only. In addition, the table of experimentally determined inhibitions was moved from the Supplementary Materials to the main body of the manuscript (Table 1 in the corrected manuscript) to make it more accessible to readers.

Reviewer Response:

I appreciate the author's response and the inclusion of Table 1 into the main manuscript is helpful. But I still have a hard time understanding how these values make any biological sense. Going back to the issue with growth kinetics. In order to get these values they were compared to a flask containing just media and the organism that was grown for 72 hours? So, in the case of P. putida, all the bacteria in this control flask would be in the process of dying or already dead. Or were they grown in some sort of continuous culture, where the media was replenished over time? What were the actual CFU counts in the control flask and experimental flasks?

I am sorry, but no matter how well fit or thought out your models are, the biological realities of your test organisms cannot be ignored for simplicity sake.

Comment 6: Use of citations

Answer:

Paper by Miloloža et al. (2022), i.e. “reference 40”, is a review paper that refers to the statement that ethylene glycol is one of the products of PET depolymerization. This review does contain reference to the depolymerization of PET (Taniguchi et al., 2019). We are sorry to hear that the Reviewer had difficulty in finding this reference. To correct this, we have replaced the Miloloža et al. reference with another reference:

Taniguchi, I.; Yoshida, S.; Hiraga, K.; Miyamoto, K.; Kimura, Y.; Oda, K. Biodegradation of PET: current status and application aspects. ACS Catal. 2019, 9, 4089–4105. DOI:10.1021/ acscatal.8b0517

Reviewer Response:

It is not the reviewer's job to go through a review article to find the original reference. The reference list of the manuscript you published is provided in order to present the original source of the research cited. It is not good scientific practice to reference a review article instead of the original source.

Also, stated above, the authors continually misrepresent the science in the citations they use. I have found three such examples above. This does not bode well for the rest of the references.

Comment 7: English

Answer:

We checked the manuscript carefully and removed the typos.

Reviewer Response:

Thank you.

Additional Reviewer Comment:

As a rule, it is not good practice to make assumptions about a reviewer's gender.
